# Behaviour, Furnishing and Vertical Space Use of Captive Callimico (*Callimico goeldii*): Implications for Welfare

**DOI:** 10.3390/ani13132147

**Published:** 2023-06-29

**Authors:** Amanda Bartlett, Lena Grinsted, Marianne Sarah Freeman

**Affiliations:** 1School of Biological Sciences, University of Portsmouth, Portsmouth PO1 2UP, UK; lena.grinsted@port.ac.uk; 2Animal Health and Welfare Research, University Centre Sparsholt, Winchester SO21 2NF, UK; marianne.freeman@sparsholt.ac.uk

**Keywords:** evidence-based welfare, callitrichid, Goeldi’s monkey, vertical space, behaviour, enclosure use

## Abstract

**Simple Summary:**

Callimico are a small primate species of conservation concern. While commonly found in zoos there is little published research relating to their captive behaviours and space use. Observational research was undertaken to address this knowledge gap by examining differences in the type and amount of behaviours in five different UK collections, and how different vertical zones of their exhibits were used. We found that there were differences in levels of behaviours between collections, including locomotion and foraging. The height use reflects their natural ecology as their behaviours were linked with the different heights within their enclosures. These findings offer evidence to support the importance of foraging enrichment such as whole foods and floor substrate to extend foraging and feeding time as well as validating EAZA’s recommendations for enclosure height for this species. The results allow a better understanding of suitable furnishing for callimico and create a springboard for further research to enhance optimum captive care for both callimico and the wider callitrichid family.

**Abstract:**

Provision of optimal captive care should be supported by species-specific evidence. Callimico (*Callimico goeldii*) is a small South American callitrichid primate. This study sought to address gaps in species-specific knowledge and captive management research by examining differences in callimico behaviour across multiple collections, investigating vertical enclosure use and a possible association between specific behaviours and vertical zones. Observational research was conducted at five European Association of Zoos and Aquaria (EAZA) organisations, in exhibits that were visually divided into four vertical zones. Instantaneous scan sampling was used to record behaviour and location of callimico over a six-day period at each collection, exceeding 160 observational hours. Significant differences were observed in foraging between collections and were much lower than the recommendations in Best Practice Guidelines, although near-wild levels were recorded in one enclosure. At an average height of 2 m, callimico utilized a similar vertical space across very different enclosures, regardless of overall available height, reflective of their natural ecology. The association between whole food items and increased foraging time, horizontal branches and locomotion and deep substrate and diversity of behaviours, offers further species-specific evidence of how the callimico use their captive environment. Our findings complement current EAZA guidelines to support species appropriate care for callimico and makes specific recommendations for further research.

## 1. Introduction

Callimico (*Callimico goeldii*), or Goeldi’s monkey, are a small South American primate of the callitrichid family that exploit a dietary niche in the dense understory of both secondary and bamboo rainforest in the Amazon basin [1,2]. Living sympatrically with red-bellied (*Saguinus labiatus*) and saddleback (*Saguinus fuscicollis*) tamarins, they face anthropogenic threat to their habitat and are highly valued within the illegal pet trade [1,3]. Their consequent listing as “vulnerable” by the International Union for Conservation of Nature (IUCN) supports their inclusion within the European Ex situ Programme (EEP), with callimico exhibited in almost 90 zoos across European Association of Zoos and Aquaria (EAZA) collections. EAZA’s Callitrichid Taxon Advisory Group (TAG) has created extensive Best Practice Guidelines to support captive care and welfare but, with 62 species of callitrichid listed, the guidelines are not species-specific [4]. Callimico are a unique genus within callitrichidae, distinct in having single births and differing dentition [4]. There is also variation in callitrichid morphology with different hand shapes associated with specialized foraging [5], while differing hindlimbs influence locomotion and substrate use [6] which is associated with differences in habitat use [2,7].

Field studies have provided knowledge of the natural ecology and in situ behaviour of callimico [2,7] that is essential to consider in the provision of appropriate captive care [8,9]. However, caution should be exercised in drawing direct comparisons between captive and wild activity budgets as environments and motivation will be very different [10]. Ex situ callimico receive regular food, lack natural predators, and have little control over social structure. In situ callimico troops of two to eleven individuals inhabit home ranges up to five times greater than sympatric red-bellied and saddleback tamarins [11]. Yearlong observations of these arboreal primates found significant variation in the vertical exploitation of the shared habitat. All species were noted to forage up to 25 m high, but this was exceptional for callimico who only briefly visited the higher vertical zones to retrieve easily accessible fruit, instead showing a clear preference for the understory. Spending the majority of their time below 3 m to 5 m [11,12] callimico also forage terrestrially, unlike the saddleback and red-bellied tamarins [11].

Research for ex situ callimico is biased towards reproductive and health matters as well as phylogeny [13,14,15] with little literature focused on species-specific captive welfare. However, a growing body of general captive research demonstrates the need to adopt an evidenced based approach to captive care [16,17] with awareness of species-specific adaptations and behaviour to develop husbandry protocol and enclosure design, and to assess welfare [18,19,20]. Understanding the species-typical behaviours is important to assist in identifying changes in activity patterns, or the appearance of abnormal, stereotypic behaviours [18,21], and can also suggest changes to enrich an animal’s environment and experience to promote species appropriate behaviours [22,23,24]. The appropriate proportions of behaviours such as foraging, social grooming and species appropriate locomotion are essential for the complex physiological and psychological needs of primates including callitrichids [4]. Limited study of captive callimico behaviours demonstrates disparity between these, the wild activity budgets and the generalised recommendations seen in the Best Practice Guidelines (Table 1).

Table 1 shows that both captive studies recorded similar foraging activity (20.6 and 19% [25,26]). Foraging has been described as functionally different from feeding by fulfilling a psychological as well as physiological need in animals [9,27]. Restricted foraging has been associated with significant welfare problems in a variety of species including psittacine [28], elephants [29] and baboons [27]. However, recommended levels of up to 37% are very different from the low levels of foraging by in situ callimico (<2%), never-theless this could be underrepresented due to the species elusive nature and dense habitat. The same study suggested foraging budgets of 10% for the saddleback and 12% for the red-bellied tamarins [2].

Maintaining behavioural fitness is essential in captive animals, particularly those held under a breeding programme. While callimico are not currently subject to a reintroduction programme, it is important that animal management preserves their natural behavioural traits as variance in behaviour can manifest in as little as a single generation [8,30,31]

Studies with chimpanzees and orangutans demonstrate that understanding vertical space use can be valuable in the provision of appropriate housing and husbandry [30,31,32]. Currently, information can be drawn from wider callitrichid research [33,34,35] but ex situ knowledge for callimico is restricted to a single study showing preference for a vertical area of 1–2 m in enclosures up to 4.9 m high [26].

Animals with wide natural ranges can be impacted negatively by captive constraints [36] and it has been suggested that no enclosure can be too large for a callitrichid [37]. The modified spread of participation index (SPI) measures how evenly an exhibit is used by looking at the actual number of behaviours observed in defined unequal zones against the number of behaviours we would expect to see if the enclosure was used equally throughout [38,39]. Not using, or actively avoiding a zone, may suggest an enclosure is unsuitable, while even use is linked with positive welfare [39]. Ross et al. [31] used SPI to examine space use of both chimpanzees (*Pan troglodytes*) and gorilla (*Gorilla gorilla gorilla*) over a four-year period and, while finding very selective use of enclosure zones, concluded that enclosure size may be less important than the relevance and functionality of the available space. Quality over quantity. Determining if there is a relationship between an area and key behaviours exhibited can offer clarity about how an enclosure is utilised and provide evidence for effective management [31,32,40].

Our study aims to address a gap in the current knowledge of captive callimico by investigating whether (i) there are differences in the frequency of behaviours across multiple captive collections, (ii) callimico use vertical space in their enclosures evenly, and whether (iii) there is an association between specific behaviours and vertical enclosure use. This knowledge will complement current Best Practice Guidelines to support species-specific assessment of welfare and provide evidence for species appropriate enclosures and guide the implementation of husbandry to promote positive captive welfare for callimico and highlight areas for further research.

## 2. Materials and Methods

### 2.1. Study Subjects and Enclosures

We conducted observations at five collections within the British and Irish Association of Zoos and Aquaria (BIAZA) that varied in height, complexity, and social structure, and with a range of callimico aged from 5 months to 27 years in a variety of exhibits styles. Both collections A and D were large, mixed sex social groups, collection B was a geriatric all male family group, and collection C and E comprised of a male and female pair. Collections A, C and D were two-part (internal and external) enclosures. Collection C and E were mixed exhibit enclosure, and both collection B and E were indoor rainforest themed enclosures with the latter a walk-through exhibit (Figure 1 and Table 2).

The callimico at collection C always had free access throughout the study to both their internal and external enclosures. As these were the same height and could be divided into comparable vertical zones the data for these two enclosures were combined. This was not possible at collection A as the external enclosure was up to double the height of the indoor enclosure so internal and external exhibits were treated separately. There was also a freely accessible area ‘off show’ in the internal enclosure where observations could not be recorded.

Observations at collection D were limited to behaviours exhibited in the external callimico enclosure. Collecting observations from the internal enclosure was compromised by the highly reflective glass and by the configuration which also included a freely accessible, regularly used, ‘off show’ area.

Each observed enclosure was visually divided into three roughly equal vertical zones using recognizable furnishings or structural elements to allow for consistency in data collection. The zones were numbered 1–3 in descending vertical height order with a fourth zone, 4, allocated to the floor of the enclosure. Measurements for exhibits were supplied by organisations when available, or otherwise estimated.

No adjustments to husbandry routines or ‘meet the monkeys’ visitor experiences were made during the data collection phases.

### 2.2. Behavioural Data Collection

Data collection took place in April, May and September 2022, when outdoor temperatures were comparable and to avoid main school holidays as far as possible. All data was collected during opening hours, which varied by collection, but typically from c9.30/10.00 a.m. to c4/5 p.m., across six separate days at each collection.

Live, real-time observations were conducted using instantaneous scan sampling at 90 s intervals over 45 min sessions, with ~15 min breaks in-between sessions [41]. From the commencement of each scan the vertical zone location and behaviour of each visible member of the group was recorded. Due to the difficulty in accurately identifying and tracking callimico, particularly in the larger enclosures, individual identification was not possible, and therefore not recorded. Where there was any doubt that individuals may have moved, particularly in the larger groups, no record was made during that scan to avoid collecting data multiple times on the same individual.

On average, 42 individual observation sessions were undertaken over a six-day period at each collection, totalling approximately 160 h of observations. An average of 1057 data entries were made per individual callimico across four of the collections, the average from collection D was much lower due to the free movement between the observed outside and unobserved inside enclosure and inability to identify each individual.

An ethogram (Table 3) was developed to identify behaviours and to prevent ‘observer drift’, ensuring consistent recording of behaviours and allowing replication of the study [42]. Abnormal behaviours were not included in the ethogram as the literature is not explicit for callimico, although any unexpected behaviours were recorded separately. The main behaviours recorded were: scanning, locomotion, feeding, foraging, self-grooming and allo-grooming. An ‘other’ category was created to condense behaviours that were very low in frequency or not observed across all collections. This included stationary behaviours where scanning did not occur such as clinging, perching and lying, and interspecific and conspecific interaction or those not identified by the ethogram. Behaviours are not exclusive so the main activity was noted, for example if a callimico is actively foraging it may still scan, but in this instance, foraging was recorded.

Additional information collected at each scan included the substrate used by an individual (to ensure consistency this was the substrate with which the hindquarters of the animal had contact) and the orientation of the substrate. Weather conditions and the outside temperature during each observation session were also noted.

Where possible, data was collected using the ZooMonitor app [44] on a Samsung Galaxy Tab S6 Lite. A Homder voice recorder enabled collection of Appendix A, including feeding times, visitor experiences or unexpected events within the scan period. For the larger groups at collections A and D, where accurate input via the app was compromised, all data was recorded using the voice recorder, which was subsequently transcribed into the Zoomonitor app. The ‘prompt’ sheet for interval data collection (Appendix A), enabled consistency in the data collected. There were limitations to observing individuals due to enclosure configuration and respecting visitor access. ‘Out of sight’ behaviour was recorded where appropriate, although following the precedent of a similar study with macaques, if location, proximity or substrate were known they were still recorded to enable as much information to be collected as possible [45].

### 2.3. Furnishings

Prior to data collection, a note was made of all furnishings in each enclosure including terrestrial substrate. At each scan, a note was made of which furnishing an individual’s hindquarters was in contact with. Although the construction of furnishings differed between collections a generic term of *platform* was applied to shelf or support structures whether made from wood or mesh. While there was foliage in many of the enclosures, either through natural growth or the provision of browse, *vegetation* was ascribed to denser areas of plant material such as long grassy areas, dense bushes, stands of bamboo or tree canopy which callimico could interact with or could obscure the individual.

### 2.4. Data Analysis

Raw data were uploaded from the ZooMonitor app to Excel or manually input in the same spreadsheet format from the voice recordings. Data was analysed using the built-in statistics packages (aov and chisq.test) in R using R Studio [46].

#### 2.4.1. Activity Budgets

One-way ANOVAs were used to determine if there were significant differences in the response variable, the frequency of each behaviour as a percentage of overall daily activity, between each collection. Where the response data violated the assumptions of normality the non-parametric alternative Kruskal-Wallis test was used. If a significance was recorded at the 95% confidence level, then a Tukey HSD *post hoc* test was run with multiple pairwise comparisons of the collections to uncover where the significant differences in frequency of behaviour occurred. Similarly, where the data was non-parametric the alternative Dunn *post hoc* test was run with a Bonferroni correction to reduce Type 1 errors to examine between which collections the significant differences lay.

#### 2.4.2. Spread of Participation

The modified spread of participation index (see below) was calculated for each collection,
(1)$$SPI=\frac{\sum \left|fo-fe\right|}{2\left(N-femin\right)}$$
where *fo* is the actual number of observations in a zone, while *fe* is the expected number of observations relative to the size of the overall enclosure (e.g., if the total number of observations was 200, and the relative size of the zone was 25% of the overall enclosure we would expect to see 50 observations in that zone). N represents the total number of observations across all zones and *fe min* refers to the smallest observation value in any zone.

The resulting SPI value is given as a figure between 0 (maximum and completely even use of an enclosure across zones) and 1 (minimum use of an enclosure; all observations recorded in single zone) [39]. A nominal depth of 0.02 m was ascribed to zone 4 of each enclosure regardless of substrate calculations available in Appendix A.

#### 2.4.3. Association between Behaviours, Zone Use and Furnishings

The categorical variables, specific behaviours and the zones in which they were performed, were tabulated for each collection. These were then subject to Chi^2^ test for association between the frequency of each behaviour and vertical zones/furnishings.

## 3. Results

### 3.1. Behaviour Budgets

Scanning was the most frequent behaviour recorded, with similar levels recorded across all collections (55–60%; Figure 2). The proportion of locomotion, feeding, foraging self- and allo-grooming were more variable across collections, and presented in detail below.

Analysis of daily data for each collection revealed that there was no significant difference in the performance of scanning as a percentage of the overall daily activity budget (*F*_(4,25)_ = 1.99, *p* = 0.13).

Analysis revealed significant difference between the collections for locomotory behaviour (Kruskal-Wallis *H*_(4)_ = 20.34, *p* < 0.001 with a Dunn *post hoc* examination highlighting the significant differences between the percentage of locomotion shown in collection E with collections C and D (Z = 3.41, *p* adj = 0.006; *Z* = 3.88, *p* adj = 0.001, respectively; Figure 2).

There was a highly significant difference between collections in the proportion of feeding behaviour by callimico (ANOVA *F*_(4,25)_ = 10.99, *p* < 0.001). A post hoc Tukey HSD showed the overall result was significantly higher at collection C, than all other collections: A (T = 3.38, *p* adj = 0.019), B (T = 3.62, *p* adj = 0.01), D (T = 6.59, *p* adj < 0.001) and E (3.87, *p* adj = 0.006), and also significantly lower at collection D than A (T = 3.22, *p* adj *=* 0.27) and B (T = 2.97, *p* adj = 0.46), with no significance difference found in any other pairings (*p* adj > 0.05) (Figure 2).

A significant difference in the proportion of foraging behaviour was found between the collections (Kruskal-Wallis *H*_(4)_ = 16.78, *p* = 0.002) with the subsequent *post hoc* Dunn test revealing significant differences between collection D with A and B (Z = 2.89, *p* adj = 0.038; Z = 3.70, *p* adj = 0.002, respectively; Figure 2).

There was no significant difference in the proportion of grooming behaviour between collections (Kruskal-Wallis *H*_(4)_ = 8.82, *p* = 0.07; Figure 2).

Analysis of allo-grooming was conducted with and without interspecific behaviour in the mixed species exhibit (collection E, namely allo-grooming with golden headed lion tamarins (*Leontopithecus chrysomelas*)). The difference between conspecific grooming and interspecific grooming at collection E can be seen in Figure 3. Significant differences in the percentage of allo-grooming were found between the collections (ANOVA *F*_(4,25)_ = 8.18, *p* < 0.001. Pairwise comparisons through a Tukey HSD *post hoc* test revealed that the significant difference was driven by collection A having higher allo-grooming levels compared to all other collections: B (T = 5.31, *p* adj *<* 0.001), C (T = 4.39, *p* adj = 0.002), D (T = 3.15, *p* adj = 0.031), E (T = 3.77, *p* adj = 0.007), even with the inclusion of interspecific grooming.

### 3.2. Spread of Participation Index

The spread of participation index (SPI) for each collection ranged from 0.53 for the external enclosure at D, to 0.42 for E, 0.29 for both B and A down to 0.28 at C. The lower figures for the latter three suggest a more even use of the enclosure (Figure 4). At each collection, the zone that had the highest proportion of observations was noted, and the mean of that zone’s height was calculated to create a ‘preferred height.’ An exception was made for E, where, although 68% of behavioural observations were made in Zone 3, the callimico rarely ventured below the ‘pathway’ made of vines and branches at the top of this zone. To present a more accurate reflection of zone use the approximate height of this feature was taken as the preferred height value. Despite the variation in enclosure height there was a marked similarity in the preferred height, with a mean height for behaviour observations across all the collections of ~2 m (Figure 4).

### 3.3. Association of Behaviour with Vertical Zones

There was a significant association between behaviours and vertical zone, pooled across all collections (χ^2^ = 2707.820, df = 15, *p* < 0.001, Figure 5). Allo-grooming was more likely to occur in higher vertical zones while foraging was more likely in lower vertical zones.

### 3.4. Association of Behaviour with Furnishings

We found a significant association between behaviours and enclosure furnishings (χ^2^ = 2154.84, df = 36, *p* < 0.001, Figure 6). Locomotion was more likely to take place on ropes or vegetation, feeding and self-grooming more likely on platforms. Scanning did not have a specific association but was less likely to take place on the floor area as opposed to foraging which was mainly associated with the floor area.

## 4. Discussion

### 4.1. Behaviours

Our study successfully addressed a knowledge gap in the behavioural budget and vertical space use of captive callimico, using a multi-institution approach. While we found a somewhat similar pattern in the overall activity budgets for callimico across the five collections, with scanning being the predominant behaviour observed, significant differences in individual behaviours (locomotion, feeding, foraging and allo-grooming) were observed between collections. Furthermore, our results differ from previous studies and EAZA’s callitrichid guidelines. The evenness of vertical space use was found with those exhibits of a smaller size, however it is important to note that other factors may also have impacted the zone use and while, uneven use was seen in the relatively taller enclosures at D and E, we found that the highest number of behaviours were performed in a similar vertical range across all collections regardless of enclosure height, with compelling evidence of an association between behaviours and the use of vertical space. These results allow us to consider implications for captive callimico husbandry and welfare below.

We recorded self-grooming and allo-grooming separately as they fulfil different needs for primates, the former described as essential maintenance while allo-grooming is viewed as a cooperative, affiliative behaviour [47]. However, previous callimico studies have not made this distinction. The combined means of both behaviours (total grooming of ~11%; ~6% self-, ~5% allo-grooming) was higher than that recorded in wild callimico at 7% [7]. While this may simply be because captive animals are easier to observe, other primate species have been noted to both self- and allo-groom more in captivity [48], and such behaviours can be a means to alleviate stress [49,50,51]. Differences in self-grooming activity between the collections were non-significant, but the highest levels recorded might have been in response to environmental noise, particularly the use of a leaf blower used during ground maintenance. Although less marked, maintenance work close to the enclosure at another zoo elicited a similar response (A.B. personal observation). Intense and unpredictable, loud, external noise can be stressful for zoo animals [49] including callitrichids [50]. While the physical appearance of callimico, such as bare patches in the pelt or unkempt, ‘spiky’ hair, can over time help to identify over- or under-grooming [52], the baseline reported in this study provides a tool to monitor self-grooming as a direct response to environmental disturbance. Interestingly, the levels of grooming in a previous mixed species study were considerably higher at 19%, perhaps suggesting that the mixed species exhibit, which was ultimately unsuccessful, was stressful for the subjects, though the difference in sampling period may also be a reason for the discrepancy [26]. The ability to exercise choice over space use can alleviate the impact of stress [53]. Callimico were noted to retreat to their internal enclosure as disturbance increased. It is also important to note that as grooming behaviours often occur out of view the proportion of time observed here could be underestimated.

Allo-grooming is essential for social cohesion between primates and contributes to parasite control [54,55,56]. In one collection with low intraspecific interactions, the female callimico did interact with the other callitrichids in the enclosure: two golden headed lion tamarins. This largely consisted of mutual grooming. When included in the allo-grooming budget, this combined behaviour exceeded the study mean of ~5% and implies positive, affiliative relationships between the callitrichid species in this mixed species exhibit [57]. This was not recorded between the pygmy marmosets and callimico in a previous study, which specifically focused on compatibility of the two species [26]. The recent introduction of a younger male into the exhibit at collection E, three weeks before the observation period began, may explain the low levels of allo-grooming involving this individual. Nonetheless, there may be welfare implications for the male, as allo-grooming is suggested to be a rewarding and pleasurable experience [47,56,58]. While the near absence of this allo-grooming may not signify negative welfare, advances in captive care research do align pleasurable experiences with positive welfare [56,59,60]. Allo-grooming may also be a mechanism to alleviate tension [37,57]. The significantly higher levels we found at collection A may relate to non-contact, aggressive, social disturbances noted from time to time within the enclosure. Limitations in identifying the individuals in the larger groups make interpretation of this more difficult.

We observed allo-grooming almost exclusively in a single zone in each one of the exhibits, all at around 2 m above ground, which may be explained by the availability of larger horizontal surfaces in these zones. Generally noted towards the rear of enclosures, or less visible to visitors, our observations suggest that the collections were meeting the needs of the callimico to find a ‘safe place’ as vigilance is reduced during allo-grooming [61]. This was evidenced elsewhere either in a high mesh tunnel outside of the enclosure, canopy vegetation as a visual barrier, or behind logs largely concealed from visitors. Multiple external platforms at the entrances between the external and internal enclosure meant allo-grooming was catered for in the larger group sizes, allowing separate groups to allo-groom simultaneously, though they were often disturbed by other members of the group entering and exiting the enclosure. When no other horizontal platforms were elsewhere in the enclosure there may have been a restriction in choice. The relationship between levels of self-grooming, allo-grooming and stress is complex. Closer examination of the effects of the social dynamics in enclosures, and the effect of the wider environment including visitors, could offer some clarity on how these affect callimico.

The significant differences that we found in levels of locomotion between collections cannot be readily explained by enclosure size. The lowest amounts of locomotion were recorded in both the largest and smallest, enclosures. The low level of locomotion recorded in a naturalistic free-ranging environment was unexpected. However, the data collection method did not record the duration of behaviours which may contribute to this. The average daily locomotion of 11.6% was less than both the locomotory budget in the wild of 17% and the 22% recorded ex situ by Dalton and Buchanan-Smith [26]. Restricted locomotion can have welfare implications. In the first instance it is related to increased body mass in captive callitrichids [57]. Chemical contraceptives can lead to weight gain, as can higher sugar diets. The provision of fruit has been linked to weight gain in captive primates [62,63]. Research into a possible relationship between physical health indicators and locomotory behaviour in callimico is therefore recommended. Aside from health factors, others such as the distribution of resources (food, platforms), complexity of branching, reflecting ease of access between resources, social pressures and stocking density may all have an impact on locomotion and general behaviour. This would benefit from further research to elucidate the key factors involved. Although our data collection did not distinguish locomotion styles, the use of the pathway of suspended branches was predominately a hopping and bounding movement prompted by long hindlimbs [64]. We observed, however, that locomotion was also associated with the display of trunk to trunk leaping which accounts for 46% of all travel in wild callimico [11]. We recommend further research to investigate locomotion style, using a method that offers an optimal opportunity to understand movement duration. This could enable evaluation of captive callimico environments to encourage wider performance of natural movement reflective of callimico morphology both for behavioural fitness and health.

Callimico are classed as geriatric above the age of 9 years [65]. The individual most often observed lying was 27 years old. Improving captive care means zoo animals are living longer, but this brings a new set of challenges in providing appropriate care [66,67]. The animal team at the collection were mindful of a deterioration in the individual and adapted husbandry accordingly, but this anomaly demonstrates how understanding changes in usual levels of locomotion can highlight potential welfare issues [52,54]. Shortly after the completion of our data collection, the individual was euthanised due to age-related complications.

Although we found a significant difference in levels of foraging behaviour between the collections, it is the disparity between these levels, which averaged at 1.6% of our activity budgets, and the levels recommended by the EAZA guidelines that merits consideration. These guidelines suggest that in situ callitrichids forage for 37% of their day. Furthermore, they are clear that it is essential to provide for this amount of foraging in a captive environment [54]. A lack of opportunities to display natural behaviours can be ‘inherently stressful’ [68] and can lead to unnatural levels of social interaction [54,69]. Maintaining natural skills is particularly pertinent for callimico who are held in a breeding program. Callitrichid parents play an active role in teaching essential problem-solving skills such as foraging to their young [70,71]. A lack of foraging opportunities can affect the behavioural fitness of callimico. The perils of not encouraging and preserving natural behaviours have been illustrated by challenges faced in reintroduction programs for callitrichids, including golden-lion tamarins (*Leontopithecus rosalia*). Inexperience in foraging and locomotory skills have compromised the success of animals in negotiating their natural environment [72]. Reported wild foraging levels in callimico at 6% of their daily budget are considerably lower than the EAZA’s general guidelines but are more relatable to the levels seen in our study.

Observations of feeding behaviour were influenced by husbandry routines and directly related to routine food provision during data collection sessions. Notable were the higher levels of feeding behaviour recorded at collection C, nearly twice that of the 6.5% average we recorded across the study. They related to the prolonged feeding on single large food items including chunks of red pepper and a whole carrot, which were taken either from the food bowls of, or, on occasion, directly from, the two-toed sloths that shared the enclosure. Vegetables were frequently served in bowls or on platforms, in relatively small, chopped pieces at all of the collections. Food presentation is a growing area of zoo nutrition research. Food should not only be biologically appropriate but also reflective of an animal’s ecology to stimulate the natural feeding and foraging behaviours that are aligned with positive animal welfare [60,73]. While entertaining for the visitors, the callimico were seen to work hard to consume the larger food items, balancing and manipulating them, much the same as would be necessary with wild sourced fruits. The provision of whole food items prolongs feeding behaviour [73] and has been found to be beneficial to a range of captive animals including parrots [28], coati [74] and primates [75]. Provision of whole foods can preserve the nutritional value of the food and minimal preparation saves valuable keeper time [63,74,76], and may promote social food sharing seen in wild callimico [11]. Hence, we suggest introducing larger or whole food items into callimico diets where nutritionally appropriate, and that further research be undertaken to better understand the effects of food presentation on feeding behaviour.

Dropped food items were also retrieved, which further reflected wild behaviours. Sympatric tamarins will take specific fruits only from the trees while callimico are noted to only retrieve them when they fall to the ground [11]. The association between foraging levels and vertical zones offers some solutions to addressing this potential low level foraging problem. Foraging noted at all collections in zones above the ground often related to the callimicos’ curiosity in exploring cracks and gaps in enclosure structures, behind shelves, and peeling bark or frayed ends off rope. The callimico also investigated unbaited enrichment devices including cardboard tubes and a log feeder. Food provision itself can be enriching [73]. Alongside the use of whole foods already suggested, we recommend smaller food items could be scattered or hidden more widely around the enclosure rather than be simply served in bowls as they currently often are. This is a simple time- and cost-efficient way of promoting a more natural way of obtaining food, with placement also able to encourage wider enclosure use. Whilst food should be accessible, a need to balance or stretch to reach it will enhance natural movement. The majority of arboreal foraging was on natural vegetation. Interaction with natural vegetation is stimulating for captive animals [77] and requires balance and manipulation. Slender bamboo stalks were particularly popular (A.B. personal observation). Even in internal enclosures the introduction of planting in hanging baskets or in large plant pots could stimulate this behaviour in a cost effective and time efficient way [68]. We suggest that plants are rotated as enthusiastic foraging can be destructive [A.B. personal obs.].

Callitrichid species employ different, and sometimes combined methods for foraging, often dictated by their morphology and natural ecology [4,5,11]. In the mixed species exhibit, the golden headed lion tamarins probed and inserted their hands into a hollow in a wooden log, while the callimico would only peer into it. The guidelines also describe grasping, tactile exploration, and ‘gleaning’ as foraging methods, with this latter method involves remaining motionless while visually inspecting the branches [4]. Ascribed to tamarins and marmoset species this approach is expanded in relation to enrichment provision, and suggests marmosets spend much of their foraging time ‘scanning’ for insects and employ a pouncing technique for catching them [68]. The guidelines say little is known of callimico foraging but they have been observed using a ‘pounce and grab’ technique [11,12] [A.B. personal obs.].

Differences in foraging techniques, as well as the disparity between the guidelines, the wild study, and the low levels of foraging recorded in our study, prompted us to reconsider how foraging behaviour presents in callimico. Whenever we observed foraging it was preceded by scanning behaviour, a distinct deliberate sweeping movement of the callimico’s head. Levels of scanning, the most prevalent behaviour recorded in wild callimico, was also consistently the most performed behaviour across the study with the average from our observations the same as the wild budget of 60%. This behaviour is not noted at such high levels in other callitrichid species including buffy-headed marmosets (*Callithrix flaviceps*) who engage in little visual scanning behaviour [78], while moustached tamarins (*Saguinus mystax*) scan for 9.7% of their daily budget [79]. Porter [7] observed the sympatric tamarins also scanned less than callimico. Scanning is largely attributed to vigilance, sometimes in response to disturbance [76], and to avoid both terrestrial and aerial predators [78,80,81,82]. We suggest an additional function of scanning should be considered: visual foraging, which has been observed in other animals such as lemurs and lorises [83,84]. Porter conceded that alongside vigilance, scanning could be used to look for food, although her field study could not clarify this [11]. While not discounting its role in vigilance, we propose that scanning forms part of the callimico foraging technique. If this is the case, then recorded levels of foraging behaviour both in the wild and in our study may be underrepresented, meaning there is less disparity with the industry guidelines than thought. We believe that this has important implications for how much this behaviour should be encouraged in callimico and given the possible complications of standard definitions of ethogram behaviours, a standardized ethogram for callimico and other callitrichids should be defined.

### 4.2. Vertical Zone Use

Callimico made the widest use of vertical enclosure space in collections A, B and C. While evenness of use implies appropriate housing, the wider use at collection A may also be explained by a density of less than 2 m^3^ per individual in the internal enclosure, which is considerably less than the other enclosures. For example, we calculated around 463 m^3^ of space per animal at collection E, including the other primates in the exhibit. The higher SPI results at collections D and E suggest that callimico do not use these larger enclosures as evenly, which could infer that certain areas are avoided or inaccessible, which could have negative welfare implications [31,39,40]. However, we suggest our findings reflect the natural ecology of callimico. At approximately 4.5 m and 8 m, both enclosures D and E exceed the in situ 3 m vertical understory range suggested by Pook and Pook [12]. Collection E also exceeds the 5 m understory range suggested by Porter [7]. When we considered the association between behaviours and vertical height, we found that Zone 1 at both collections, which represented areas of the enclosures above their natural vertical range, were rarely used.

As seen in great apes, SPI can be a useful indicator, but we suggest it should be used with knowledge of an animal’s natural environment to understand the relevance of zones [31,32]. Although direct comparison between the vertical zones of the exhibits is not possible due to the disparity of height, the identification of a favoured vertical zone in each enclosure offers an important insight into the vertical space use of captive callimico. At all collections, despite furnishing opportunities at greater heights, we observed that between 50% and 70% of behaviours were recorded at an average height of 2 m, well within their natural vertical range. This has important implications for enclosure design. Although there is merit in providing captive callimico the opportunity to explore a higher vertical range as their wild counterparts occasionally do, we established that EAZA’s minimum recommended enclosure height for callitrichids of 2.5 m [50] allows callimico to exhibit a full range of natural behaviour in a way that reflects their natural ecology. Resources can be targeted to offer greater enclosure complexity at this understory level. Understanding this vertical space use can support the creation of mixed species exhibits with animals that are more likely to use the higher reaches of an enclosure. However, our results do pose a concern that the recommended height may be less suited to callitrichid species who exploit a higher natural range. Further research is therefore needed to improve our understanding of how other callitrichid species use their captive vertical space.

Our investigation found a significant association at each collection between the observed behaviours and the vertical zone they were recorded in. For some activity this could be explained by the features of the enclosure, as seen with allo-grooming or husbandry routines, observed with feeding. Scanning and locomotory behaviours revealed similar patterns of vertical zone use with the highest levels occurring in zones with fixed ‘pathways’, made from rope or logs joined to make a track. Although these horizontal pathways may be placed to benefit visitor viewing, callimico were recorded to use horizontal substrate in 54% of observations in the wild [11]. We should be mindful that the natural ecology for callimico is within a seasonally changing environment. Regular, complete changes of enclosures would be time consuming for zookeepers and stressful for callimico [50]. We suggest that enclosures can be easily and cheaply modified by occasionally moving ropes or branches to provide unpredictability, encourage navigational behaviour and offer some choice and control within a restricted environment [50,85,86]. Furthermore, singular, regularly used pathways may make identification of abnormal repetitive locomotory behaviour more difficult to identify, which has implications for assessing welfare. Literature is limited for callimico but callitrichids are reported to route trace in response to stress [87].

The clearest difference seen across the collections was the association of terrestrial foraging behaviour at some collections, where we observed deep bark chip and loose soil substrate encouraging interaction not seen on the concrete or compacted soil of the other collections. Brief, ‘pounce and grab’ behaviour was seen in all exhibits, but loose substrate also promoted manipulating bark, raking soil, and turning fallen leaves to retrieve fallen food or naturally occurring insects described in wild callimico. A lack of success did not appear to deter this behaviour. A reluctance to forage in densely planted areas, even for high value food items, was observed, which may reflect the reluctance seen in the wild [88] to insert hands into unknown areas, and the inability to scan the ground here may well offer further support to our proposition that this is also part of the foraging repertoire. We recommend evaluation of terrestrial substrate use in callimico enclosures as an effective way to promote natural foraging behaviour and also to promote wider enclosure use to lower levels of exhibits. A caveat to this recommendation is the acknowledgement that zoo staff need to be mindful of pests that may be harmful to the animals in their care. The introduction of soil in enclosures has caused cockroach issues previously at one of the collections. The substrate may have come from an unsuitable source, but interestingly a similar problem was countered at another EAZA zoo by the successful introduction of smooth sided toad (*Rhaebo guttatus*) [89].

### 4.3. Future Directions

Our use of the ZooMonitor app and repeatability of the method lends itself to an expansion of this study. The opportunity to collate data from a much larger sample of captive environments would offer further clarity to behavioural differences across collections, strengthen the evidence of vertical space use and may reduce the significant differences noted in behaviour to enable a baseline, that follows a standardised ethogram, to be developed for welfare assessment. It would also be interesting, following on from this study, to determine whether there is a change to the use of vertical space depending on the number of visitors or other environmental disturbance; it is possible that higher areas of the enclosure are necessary to allow space to retreat.

Expanding this study to other callitrichid species would provide evidence about where similarities and differences lie between the species to enhance species-appropriate guidance and support the use of mixed species exhibits. The inclusion of multiple callitrichid species in the further research recommended in our study would strengthen our understanding of interspecies differences.

## 5. Conclusions

The opportunity to consider callimico behaviours and enclosure use, across multiple environments, has succeeded in addressing a knowledge gap in the captive activity budgets and vertical space use of this species. Differences in observed levels of behaviour highlighted areas that warrant further consideration due to potential welfare implications, including grooming and locomotion. Observations of foraging levels were much lower than those recommended by EAZA’s callitrichid guidelines, although near wild levels were recorded in one enclosure. We recommend collections to adopt similar foraging enrichment as this enclosure, providing whole foods and woodchip on the floor that both extends foraging and feeding time. Consideration was prompted about whether scanning behaviour is part of the callimico foraging repertoire, which could mitigate the difference with the recommended levels. Callimico were found to utilise a similar vertical space across very different enclosures, reflective of their natural ecology. This validates EAZA’s recommendations for enclosure height for this species and provides evidence for enclosure design and the placement of furnishings and enrichment. The association of behaviours with vertical space offers further species-specific evidence of how the callimico use their captive environment, with a preference for 2 m above ground and rare use of higher levels. Our findings complement current Best Practice Guidelines and provide evidence to support species appropriate enclosures and husbandry to promote positive welfare for callimico. Research recommended by our findings and understanding how our results translate to other callitrichid species will continue to enhance our knowledge to support the provision of species appropriate captive welfare of this enigmatic callitrichid.

## Figures and Tables

**Figure 1 animals-13-02147-f001:**
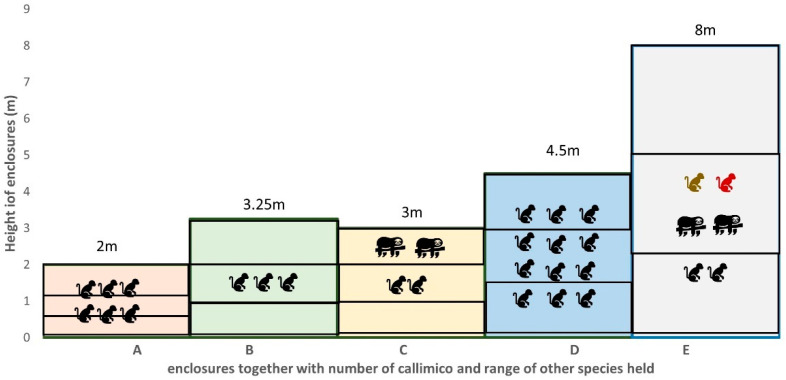
Representation of enclosures by height together with number of callimico and a range of other species mixed within each exhibit. Horizontal lines in each bar represent the division of height zones in each enclosure. Enclosure size and height varied greatly, E, at 2780 m^3^, was over 240 times larger than enclosure A at 11.5 m^3^. Disparity in animal density can also be seen. Even when accounting for all the primates housed in E (*n* = 6) the space per individual at 463 m^3^ is 14 times larger than the European Association of Zoos and Aquaria’s (EAZA) recommended enclosure size of 32 m^3^ for up to five callitrichids.

**Figure 2 animals-13-02147-f002:**
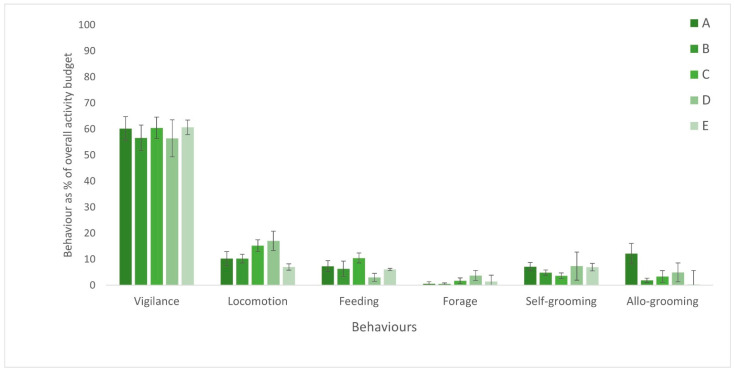
Overall activity budget from each location with each behaviour as an average of the total daily percentages at that collection. Error bars show standard deviation for each behaviour across the six-day study period. Bars represent each collection A–E.

**Figure 3 animals-13-02147-f003:**
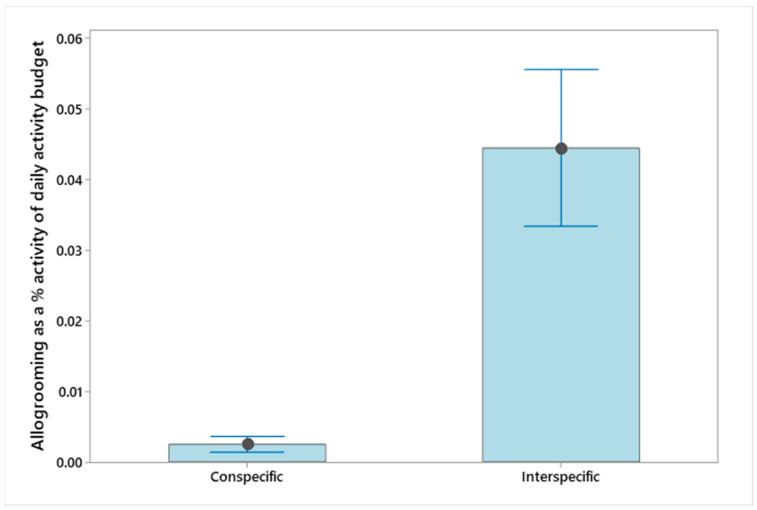
Conspecific and inter-specific allo-grooming as a percentage of activity budget at collection E. Error bars represent standard error.

**Figure 4 animals-13-02147-f004:**
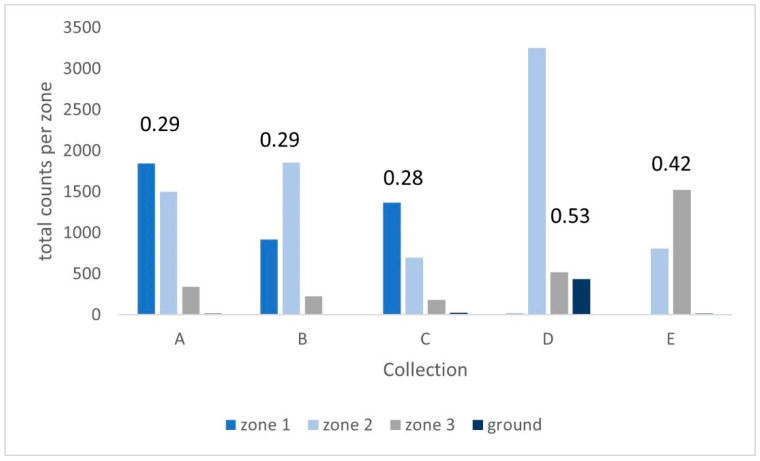
Zone use and SPI values for each collection. Note that SPI values run from of 0 to 1 with a lower score indicating wider and more even use of an enclosure. The external enclosure at collection D sees the least even use.

**Figure 5 animals-13-02147-f005:**
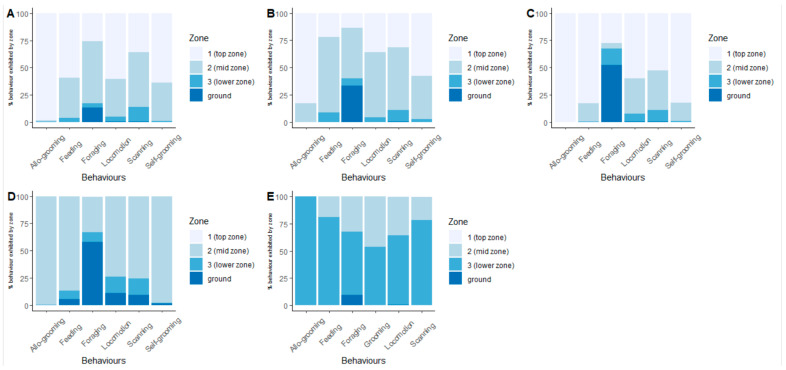
Association of behaviours with vertical zones at each collection. Each bar represents the total observations of a specific behaviour over six days. The colours represent the percentage of time that behaviour was observed in that zone. Note that for E the majority of Zone 2 exceeds the vertical range of 3 m for wild callimico described by Pook and Pook [12]. Zone 1 also exceeds the vertical range for wild callimico described by Porter [2]. Graphs represent each collection (**A**–**E**).

**Figure 6 animals-13-02147-f006:**
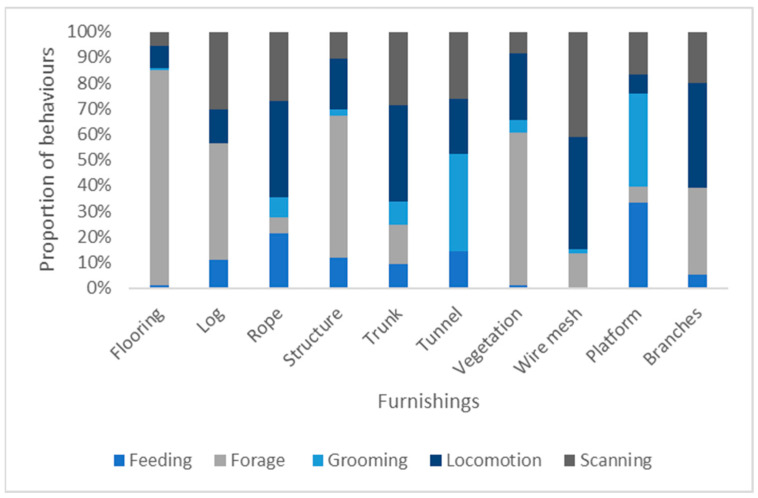
Proportion of behaviours occurring at different furnishings across the different collections.

**Table 1 animals-13-02147-t001:** Disparity in published callimico activity budgets.

	Activity during Feeding Enrichment, Ex-Situ [25]	Mixed Species Exhibit Study, Ex-Situ [26]	In Situ Field Study [11]	General Callitrichid Ex-Situ ‘Best Practice’ Guidelines [4]
Locomotion	23.3%	22%	17%	
Foraging	20.6% *	19%	6%	Up to 37%
Grooming		27%	7%	
Scanning			60%	

* Combined feeding and foraging value.

**Table 2 animals-13-02147-t002:** Summary of the five collections and study subjects.

Collection	Observation Dates	Study Subjects	Enclosure
A	17 May 2022–22 May 2022	♀ 16y.6m ♂ 6y.11m ♀ 6y.6m ♂ 5y.11m ♂ 5y.0m ♂ 4y.6m	Open fronted external enclosure with moat and glass fronted internal enclosure. Internal enclosure 11.5 m^3^/external enclosure 100 m^3^
B	4 September 22–9 September 22	♂ 27y.2m ♂ 21y.4m ♂ 18y.1m	Enclosure is part of a glasshouse rainforest exhibit; 51.19 m^3^
C	12 September 22–17 September 22	♀ 1y.10m ♂ 8y.2m	Enclosure shared with pair of southern two-toed sloth (*Choloepus didactylus*); 136 m^3^
D	9 May 2022–14 May 2022	♀ 9y.11m ♂ 3y.0m ♀ 3y.0m ♂ 2y.6m ♀ 5y.1m ♂ 2y.6m ♀ 1y.4m ♂ 1y.10m ? 0.10m ♂ 1y.6m ? 0.4m ♂ 0.11m	large planted outside area and glass fronted internal area; 67.5 m^3^
E	18 April 2022–23 April 2022	♀ 18y.9m (nonbreeding) ♂ 9y.11m (introduced 3 weeks before observations)	Rainforest themed ‘walk-through’ enclosure with multiple South American mixed species (including red titi monkeys (*Callicebus cupreus*), golden headed lion tamarin (*Leontopithecus chrysomelas*), southern two toed sloths (*Choloepus didactylus*) and tamandua (*Tamandua tetradactylac*)); 2780 m^3^

**Table 3 animals-13-02147-t003:** Ethogram used for behavioural data collection, drawn from both species-specific and wider callitrichid sources [33,37,43].

Behaviours	Definitions
Clinging	Subject is in a fixed position within the enclosure, holding on with claws to a vertical surface. There may be head movement but not ‘scanning’.
Perching	Subject is in a fixed position on a largely horizontal surface which may include but is not limited to shelving, nest boxes, ropes and branches. There may be head movement, but not ‘scanning’.
Lying	Subject is spread across a surface—can be facing downwards or on side—or may have hind legs tucked under—no scanning movement of the head
Scanning	Involves subject adopting a still posture while moving the head in a distinct, vigilant, motion as if monitoring the area. This movement is often directed downwards but may be observed with the head tilted upwards.
Grooming	Self-grooming which may include but is not limited to the subject scratching themselves with fore or rear claws or rubbing themselves against bark other surface.
Allo-grooming	Grooming occurring between subjects including but not limited to one subject manipulating or picking through or the fur of the other.
Conspecific interaction	Any contact between subjects that does not constitute grooming. It may involve but is not limited to play, aggression, food sharing or sexual activity.
Allospecific	Any contact between callimico and other animals within the enclosure (for mixed species exhibits).
Locomotion	Movement around the enclosure by the subject using a leaping, bounding or clinging motion and can include, but is not restricted to, passage across shelving, wire, panels, branches and ropes.
Foraging	The active seeking of food by the subject which may include the manipulation of a substrate but does not include the consumption of food or the simple act of lifting food from a bowl/platform—but can include actively sorting through food bowl contents.
Feeding	The active consumption of food.
Out of Sight	Subject cannot be observed by the researcher.
Other	Any activity not expressly noted in the ethogram which include drinking, vocalisation or interaction with any species in an adjoining enclosure.

## Data Availability

Data can be provided by the corresponding author upon reasonable request.

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
