# Peer review of "Behaviour, Furnishing and Vertical Space Use of Captive Callimico (Callimico goeldii): Implications for Welfare"

_animals, 2023, doi:10.3390/ani13132147_

Round 1
Reviewer 1 Report
Authors characterized the difference in behavioral activity and use of vertical space in callimico in comparison to other captive studies, a study of in situ conspecifics, and the captive care guidelines for the species.
The ideas and intention behind the project are important to understanding the responses of animals to multiple aspects of human care, which Is varied, multimodal, highly species specific, and therefore necessitates further studies like this.
Overall, the manuscript is well written, though remains unedited in several locations. The methods employed were appropriate for a condensed, multi-institutional study, however, did not account for the variety of environments or management styles the animals lived in, and so do not provide a clear understanding of what factors may influence the outcomes observed. Likewise, additional information on the ethogram is needed to be able to understand and interpret the results more clearly. Overall, statistical analysis were appropriate and well-explained, the authors are careful not to overstep results in their assessment, and conclusions were clearly drawn from outcomes they received and supported by cited literature.
My comments are summarized below.
Introduction
Lns 49-67. Good summary of the setting events to the argument, and for the point of caution which further justifies the importance of this work, e.g., the guidelines are generalized, but species morphology and home range in comparison to sympatric species indicates either niche specialization or at the least differentiation. Authors loosely address this in the remainder of the Introduction and in the Discussion, but I’d recommend being more direct, where warranted.
Ln 70-71: its unclear if the 'growing body of research' refers to the previous sentence (reproductive, health, and phylogeny) demonstrating need for the following topics, or a general body of information on captive species does. Recommend clarifying.
Lns 73-78: There's a lot of concepts condensed into these sentences, which either need to be expanded on (for those less familiar with the pitfalls of activity budgets) or removed (to not be distracting to those familiar and may seek more clarity in how the concepts will be further defined in relation to this research). I would recommend revising the paragraph overall to not focus on activity budgets, but justifying the proportion of foraging, social grooming, and species appropriate locomotion that may be unique to callimico, particularly in relation to sympatric callitrichids, and are therefore essential in promoting/informing their captive experience and care.
Lns 83-87: These sentences seem to be missing a direct connection to one another; likewise, they also seem premature in the paragraph, since the first sentence and last two sentences (lns 87-90) describe the table, with these sentences noting the context and implications of comparative captive-wild studies. Consider revising.
Lns 87-89: I believe this is a sentence fragment.
Lns 83-99: These sentences and concepts read as disconnected. Recommend transitional sentences to tie concepts and sentences together to improve flow and understanding of the information presented.
Methods
Ln 130: define mixed exhibit enclosure.
Fig 1. Recommend using patterns or high contrasting colors for Enclosures labels A-E, or labeling them above the bars within the figure, since these colors may not be able to be differentiated in black and white or for people with visual impairments.
Ln 160. Appears multiple observers collected data, but I didn’t see mention of reliability being conducted. Please clarify.
Ln 164. Was data collected live, or via recorded video? Please clarify.
Ln 176: does "no identification of individuals" mean no animals were observed (because they were inside), or a separate clause in that the animals were not able to be individually recognized? Please clarify.
Ln 187. Is the ethogram listed in hierarchical order, then? Please clarify.
Ln 189: If this second example is necessary, I'd recommend using two different behaviors, or where scanning would be included over another behavior to further clarify the example above.
Ln 193-194. Why was 1m chosen? Is it biologically relevant, or for comparison to other in situ or captive studies, care standards, etc?
Ln 201. Assume that the voice recorder data was then transcribed into the app? Please clarify.
Ln197, Lns 218-220. Please confirm formatting for this Journal re. appropriate citation for technology; typically requires manufacturer, city, and company.
Lns 223-224. Does frequency data (sum to 100% of "overall daily activity") compromise the interdependence condition for ANOVA? Please clarify.
Results
Lns 296-298. This highlights a concern I've had regarding enclosure height as a metric of enclosure use, as the total cubic space in many captive enclosures is not fully accessible or functional to arboreal species, especially non-flighted mammals. I'm happy to see this adjustment was made to reflect functional space, and recommend a sentence or two higher up in the manuscript, where appropriate.
Lns 300-302. This is a great outcome!
Discussion
Lns 345-399. It is important to note that observations of grooming behaviors occurred only during the zoo's operating hours, and as authors note tend to be conducted out of view of people (ex and in situ), so necessarily representative of the full amount of time the species likely engage in the behavior. This may only be important for grooming as its assumed other behaviors described have the stimulus present to engage in the behavior during operating hours (feeding, foraging) or are less secretive (e.g., locomotion, scanning).
Lns 374-379. This example is hard to follow. I assume the male referred to in the latter two sentences is the callimico in Ln 374?
Lns 400-416.The discussion on locomotion focuses on health implications that may or may not be related to locomotion. I recommend revising this paragraph to speak towards why there are apparent differences between settings (also I’m assuming “settings” refers to in situ and captive). Though authors may not have measured other variables in enclosure design, function, and management, this may be a good opportunity to posit other variables that may be playing into the differences seen: distribution of resources (food, platforms), complexity of perching (ease of access between resources), social pressures and density, etc, or bring Lns 529-553 to follow this paragraph.
Lns 417-424. This is an interesting anecdote, but it’s unclear how it relates to the research and am concerned it distracts from the flow. Consider removing.
Lns 439-440. This is an important statement that doesn't seem to be capitalized on - whole fruit is a great insight to the differences in feeding duration, but this statement directly relates to Lns 433-435 re. alignment with the species biology and ecology. Consider elaborating.
Lns 475-495. This is a good section. To expand, how much of the disparity here is in definition of foraging and locomotion? Foraging is not exclusive to locomotion, and so these outcomes can be mixed between hierarchal ethograms and those that are more exclusive. Please clarify.
Ln 488. Please continue this sentence.
Lns 340-341, assert that uneven use of vertical is attributed to the relatively taller enclosure height for D and E, and here in Lns 498 that evenness of use implies appropriate housing, and so suggests that the housing in D and E is inappropriate on account of the enclosures being too tall. I worry it is easy to draw the conclusion from these statements that too tall an enclosure can have a negative impact on the welfare of the callimico. This may well be the case, but further observations are necessary to adequately support such a conclusion. Likewise, it seems premature to suggest that the uneven use of vertical space in D and E should be attributed to the enclosure height, given the range of factors that might have contributed to such a difference (e.g., difference in group dynamics, difference in individual personalities, difference in anthropogenic disturbances, etc.). I appreciate authors are careful to not exceed the outcomes of their study, and would recommend further review of this part of the discussion.
Lns 511-512. Excellent statements, thank you.
Lns 515-517. Since most caretakers can easily reach 2-2.5m, I assume most resources are placed within this range. A statement noting how much availability there was to use higher and lower perching, platforms, or vegetation would be appreciated. The manuscript loosely describes this aspect, but I recommend a more direct statement that clarifies.
Lns 554-589. I recommend these paragraphs be moved to be included in the Feeding section above (Lns 447-495). I removed a comment in the above requesting inclusion of this information, so am happy to see it in the manuscript, but seems to be less related to vertical use than activity budget. Alternately, could include this in its own subheading to differentiate it the sections above. This section also has great insights to the behaviors observed and possible links to what has been seen in wild conspecifics.
Lns 591-600. Likewise, I would encourage authors to mention the possible complications of standard definitions of behaviors in ethograms as they relate to the behavior observed in callimico and callitrichids, and so defining a standard ethogram for the species or taxon would aid in the sentiments expressed in this paragraph.
None
Author Response
Thank you for reading our manuscript and your helpful comments, which we have addressed point by point below.
Introduction
Lns 49-67. Good summary of the setting events to the argument, and for the point of caution which further justifies the importance of this work, e.g., the guidelines are generalized, but species morphology and home range in comparison to sympatric species indicates either niche specialization or at the least differentiation. Authors loosely address this in the remainder of the Introduction and in the Discussion, but I’d recommend being more direct, where warranted.
We have made some changes to the writing to be more direct at points.
Ln 70-71: its unclear if the 'growing body of research' refers to the previous sentence (reproductive, health, and phylogeny) demonstrating need for the following topics, or a general body of information on captive species does. Recommend clarifying.
Clarified
Lns 73-78: There's a lot of concepts condensed into these sentences, which either need to be expanded on (for those less familiar with the pitfalls of activity budgets) or removed (to not be distracting to those familiar and may seek more clarity in how the concepts will be further defined in relation to this research). I would recommend revising the paragraph overall to not focus on activity budgets, but justifying the proportion of foraging, social grooming, and species appropriate locomotion that may be unique to callimico, particularly in relation to sympatric callitrichids, and are therefore essential in promoting/informing their captive experience and care.
This has been revised
Lns 83-87: These sentences seem to be missing a direct connection to one another; likewise, they also seem premature in the paragraph, since the first sentence and last two sentences (lns 87-90) describe the table, with these sentences noting the context and implications of comparative captive-wild studies. Consider revising.
Lns 87-89: I believe this is a sentence fragment.
Lns 83-99: These sentences and concepts read as disconnected. Recommend transitional sentences to tie concepts and sentences together to improve flow and understanding of the information presented.
All three points above have been addressed with a revised paragraph
Methods
Ln 130: define mixed exhibit enclosure.
This has been clarified within the addition of table 2
Fig 1. Recommend using patterns or high contrasting colors for Enclosures labels A-E, or labeling them above the bars within the figure, since these colors may not be able to be differentiated in black and white or for people with visual impairments.
Ln 160. Appears multiple observers collected data, but I didn’t see mention of reliability being conducted. Please clarify.
There was only one observer for the study, this is clarified in text and with the addition of author roles at the end.
Ln 164. Was data collected live, or via recorded video? Please clarify.
Data were collected live- this is now clarified in the methods
Ln 176: does "no identification of individuals" mean no animals were observed (because they were inside), or a separate clause in that the animals were not able to be individually recognized? Please clarify.
The latter, we have clarified in text
Ln 187. Is the ethogram listed in hierarchical order, then? Please clarify.
Yes functional inference is made from the specific behaviours to broad categories, a more specific ethogram that was used has been included in the main paper.
Ln 189: If this second example is necessary, I'd recommend using two different behaviors, or where scanning would be included over another behavior to further clarify the example above.
Second example is not necessary and has been removed.
Ln 193-194. Why was 1m chosen? Is it biologically relevant, or for comparison to other in situ or captive studies, care standards, etc?
This sentence has been removed as we are not investigating proximity in this paper.
Ln 201. Assume that the voice recorder data was then transcribed into the app? Please clarify.
Yes clarified in text
Ln197, Lns 218-220. Please confirm formatting for this Journal re. appropriate citation for technology; typically requires manufacturer, city, and company.
We have cited and referenced as per the journal and manufacturers website but will check this a final review stage with the journal.
Lns 223-224. Does frequency data (sum to 100% of "overall daily activity") compromise the interdependence condition for ANOVA? Please clarify.
As the primary factor of interest was the collections which are all independent from each other we believe this meets the assumptions of a one way ANOVA.
Results
Lns 296-298. This highlights a concern I've had regarding enclosure height as a metric of enclosure use, as the total cubic space in many captive enclosures is not fully accessible or functional to arboreal species, especially non-flighted mammals. I'm happy to see this adjustment was made to reflect functional space, and recommend a sentence or two higher up in the manuscript, where appropriate.
We agree with your point and have ensured that discussions on functional space are included early in the manuscript.
Lns 300-302. This is a great outcome!
Thank you.
Discussion
Lns 345-399. It is important to note that observations of grooming behaviors occurred only during the zoo's operating hours, and as authors note tend to be conducted out of view of people (ex and in situ), so necessarily representative of the full amount of time the species likely engage in the behavior. This may only be important for grooming as its assumed other behaviors described have the stimulus present to engage in the behavior during operating hours (feeding, foraging) or are less secretive (e.g., locomotion, scanning).
Added this in as a caveat.
Lns 374-379. This example is hard to follow. I assume the male referred to in the latter two sentences is the callimico in Ln 374?
Yes this has been clarified
Lns 400-416.The discussion on locomotion focuses on health implications that may or may not be related to locomotion. I recommend revising this paragraph to speak towards why there are apparent differences between settings (also I’m assuming “settings” refers to in situ and captive). Though authors may not have measured other variables in enclosure design, function, and management, this may be a good opportunity to posit other variables that may be playing into the differences seen: distribution of resources (food, platforms), complexity of perching (ease of access between resources), social pressures and density, etc, or bring Lns 529-553 to follow this paragraph.
Added a note here
Lns 417-424. This is an interesting anecdote, but it’s unclear how it relates to the research and am concerned it distracts from the flow. Consider removing.
Removed
Lns 439-440. This is an important statement that doesn't seem to be capitalized on - whole fruit is a great insight to the differences in feeding duration, but this statement directly relates to Lns 433-435 re. alignment with the species biology and ecology. Consider elaborating.
Moved paragraphs to this section to help elaborate on this point
Lns 475-495. This is a good section. To expand, how much of the disparity here is in definition of foraging and locomotion? Foraging is not exclusive to locomotion, and so these outcomes can be mixed between hierarchal ethograms and those that are more exclusive. Please clarify.
Thank you, we have added a little to reflect the potential issues with sampling but also limited to be concise.
Ln 488. Please continue this sentence.
Amended
Lns 340-341, assert that uneven use of vertical is attributed to the relatively taller enclosure height for D and E, and here in Lns 498 that evenness of use implies appropriate housing, and so suggests that the housing in D and E is inappropriate on account of the enclosures being too tall. I worry it is easy to draw the conclusion from these statements that too tall an enclosure can have a negative impact on the welfare of the callimico. This may well be the case, but further observations are necessary to adequately support such a conclusion. Likewise, it seems premature to suggest that the uneven use of vertical space in D and E should be attributed to the enclosure height, given the range of factors that might have contributed to such a difference (e.g., difference in group dynamics, difference in individual personalities, difference in anthropogenic disturbances, etc.). I appreciate authors are careful to not exceed the outcomes of their study, and would recommend further review of this part of the discussion.
We have amended this section, with notes on other factors and cautionary interpretation.
Lns 511-512. Excellent statements, thank you.
Lns 515-517. Since most caretakers can easily reach 2-2.5m, I assume most resources are placed within this range. A statement noting how much availability there was to use higher and lower perching, platforms, or vegetation would be appreciated. The manuscript loosely describes this aspect, but I recommend a more direct statement that clarifies.
Added
Lns 554-589. I recommend these paragraphs be moved to be included in the Feeding section above (Lns 447-495). I removed a comment in the above requesting inclusion of this information, so am happy to see it in the manuscript, but seems to be less related to vertical use than activity budget. Alternately, could include this in its own subheading to differentiate it the sections above. This section also has great insights to the behaviors observed and possible links to what has been seen in wild conspecifics.
Restructured as suggested
Lns 591-600. Likewise, I would encourage authors to mention the possible complications of standard definitions of behaviors in ethograms as they relate to the behavior observed in callimico and callitrichids, and so defining a standard ethogram for the species or taxon would aid in the sentiments expressed in this paragraph
A very good point and we have taken the opportunity to include this here and further up in the discussion to make this clear.
Reviewer 2 Report
Congratulations on a very well considered paper. This is a novel, interesting and necessary application of well-established behavioural methods which provides an evidence basis for zoo husbandry. The discussion is extremely well configured with a detailed interpretation of the results that is balanced and substantiated by comprehensive coverage of the literature.
There are minor corrections relating to text or presentation:
Line 13-14 ‘The height use reflects their natural ecology their behaviours were linked with the different heights within their enclosures’ should read ‘…..as their behaviours were linked ……’
Line 26 should read ‘Instantaneous scan sampling was used to record…..’
Line 31 please explain further ‘the association found……’ what specifically are the associations?
Keywords – suggest adding ‘enclosure use’.
Line 85 [9,27] need a space between , and 2
Line 160 please name the months of data collection
Line 187 – prioritising one behaviour to be recorded per sample interval, opportunity for bias to creep in here as the researcher determines what is the ‘main’ behaviour, why not record all behaviours in a wider ethogram, for the example given, ‘feeding’, ‘scan’, and ‘scan while feeding’ could be recorded. This needs to be considered as a sampling bias in the Discussion.
Line 218 ‘Raw data was….’ Should read ‘Raw data were…..’
2.4 Data analysis – in this section can you please cite the R package or code used.
Section 2.4.2 can you clarify if you are using SPI or mSPI – mSPI is discussed in the Introduction, SPI is discussed in section 2.4.2
Figure 2 should not have a grey line around it and in the caption the key A-E is not explained. Also x axis label ‘Behaviours’ should be brought further down and away from the behaviour labels. A y axis line should be added.
Line 249 replace ‘dominant’ with ‘frequent’ for a more accurate reflection of the data.
Subsections 3.1.1 – 3.1.6 should be amalgamated into a single subsection.
For ANOVA results, df and N should be subscripted and p should be lower case p.
Figure 4 – how does the reader know which are the inter and which are the intra species allo-grooming results? The figure caption ends with ,. Hence delete the ,
Line 294 – refers to Figure 11 – there isn’t a Figure 11 displayed.
Figure 5 is difficult to understand. Please review caption. What are the xs for? What do the bars show? What do the dots show? There is a lot going on in this figure – I suggest removing SPI value as
I would like to see a frequency plot for zone use in each enclosure plotted along with SPI value. This would help the reader understand which of the zones were preferred.
Figure 6 – pale grey line to the left of the figure – please delete.
Line 327-328 foraging typically only on the floor – was all the food presented on the floor, effectively forcing the primates to the floor to collect food or was food presented throughout all the hight zones? This should be analysed in the Discussion.
Figure 7 – grey line around figure – please delete
Line 410 and discussion on the primates being overweight – were these primates also in the smallest cages or at the highest stocking density? Please state whether this is or isn’t also a likely cause of excess weight.
Line 488 ‘such as…’ dots need to be removed and species names inserted
Line 527 and above – it would be interesting to know if the primates use the higher enclosure zones when visitor numbers are high, or visitor noise is high. Please state this as a potential future research stream – as it may be essential to have taller enclosures to allow flee responses given the artificiality of captivity and this has not yet been studied for this species.
Author Response
Thank you for your comments, we have addressed each point below.
Line 13-14 ‘The height use reflects their natural ecology their behaviours were linked with the different heights within their enclosures’ should read ‘…..as their behaviours were linked ……’
Thank your this has been amended
Line 26 should read ‘Instantaneous scan sampling was used to record…..’
edited
Line 31 please explain further ‘the association found……’ what specifically are the associations?
The abstract has been updated to clarify this
Keywords – suggest adding ‘enclosure use’.
Thank you, this has been added
Line 85 [9,27] need a space between , and 2
Thank you
Line 160 please name the months of data collection
Included
Line 187 – prioritising one behaviour to be recorded per sample interval, opportunity for bias to creep in here as the researcher determines what is the ‘main’ behaviour, why not record all behaviours in a wider ethogram, for the example given, ‘feeding’, ‘scan’, and ‘scan while feeding’ could be recorded. This needs to be considered as a sampling bias in the Discussion.
We have added in a line to reflect issues in sampling bias and the need for standardised ethograms
Line 218 ‘Raw data was….’ Should read ‘Raw data were…..’
Amended
2.4 Data analysis – in this section can you please cite the R package or code used.
The basic in-built r packages were used- we have included the syntax of the main codes.
Section 2.4.2 can you clarify if you are using SPI or mSPI – mSPI is discussed in the Introduction, SPI is discussed in section 2.4.2
As we have uneven zones based on the ground and zone splits this would be mSPI
Figure 2 should not have a grey line around it and in the caption the key A-E is not explained. Also x axis label ‘Behaviours’ should be brought further down and away from the behaviour labels. A y axis line should be added.
We have explained the Key A-E in the caption and will correct all figures in the final edit with the high resolution images.
Line 249 replace ‘dominant’ with ‘frequent’ for a more accurate reflection of the data.
edited
Subsections 3.1.1 – 3.1.6 should be amalgamated into a single subsection.
Adjusted as requested
For ANOVA results, df and N should be subscripted and p should be lower case p.
edited
Figure 4 – how does the reader know which are the inter and which are the intra species allo-grooming results? The figure caption ends with ,. Hence delete the ,
edited
Line 294 – refers to Figure 11 – there isn’t a Figure 11 displayed.
Edited to figure 5
Figure 5 is difficult to understand. Please review caption. What are the xs for? What do the bars show? What do the dots show? There is a lot going on in this figure – I suggest removing SPI value as
I would like to see a frequency plot for zone use in each enclosure plotted along with SPI value. This would help the reader understand which of the zones were preferred.
We have altered this to a frequency plot of zone use
Figure 6 – pale grey line to the left of the figure – please delete.
Will be corrected with high resolution images
Line 327-328 foraging typically only on the floor – was all the food presented on the floor, effectively forcing the primates to the floor to collect food or was food presented throughout all the hight zones? This should be analysed in the Discussion.
The association with the floor was not because food was normally distributed this way. They either retrieved food that was dropped - originally sourced from bowls - or through the active manipulation of the substrate (bark chip/loose soil). We have added a note to this in the discussion
Figure 7 – grey line around figure – please delete
Will be corrected with high resolution images
Line 410 and discussion on the primates being overweight – were these primates also in the smallest cages or at the highest stocking density? Please state whether this is or isn’t also a likely cause of excess weight.
We have removed this section here based on the comments of other reviewers
Line 488 ‘such as…’ dots need to be removed and species names inserted
Completed
Line 527 and above – it would be interesting to know if the primates use the higher enclosure zones when visitor numbers are high, or visitor noise is high. Please state this as a potential future research stream – as it may be essential to have taller enclosures to allow flee responses given the artificiality of captivity and this has not yet been studied for this species.
A good point, we have added this in as a potential future research idea.
Reviewer 3 Report
This paper describes a study of the behaviour and vertical space utilisation of Callimico/Goeldi’s monkey (Callimico goeldii) conducted at 5 different EAZA zoos in the UK. The paper addresses a gap in knowledge about Callimico behaviour and enclosure/exhibit utilisation in captivity. The goals of the study were to assess 1) behavioural differences across the 5 zoos, 2) utilisation of vertical space within enclosures, and 3) whether specific behaviours were associated with certain vertical space utilisation. The authors provide a useful overview of existing research conducted wild and captive Callimico and related species. The behavioural methodology and data analyses were well-written and clearly described. The use of “spread of participation” metric was useful for comparing to previous space utilization studies. The results and discussion clearly summarize the findings and implications for each of the proposed study goals. The primary concerns about the manuscript pertain to the lack of any reported intra- or inter-observer reliability data, the inability to identify individual animals and the significant variation across the 5 zoos in terms of enclosure height/design, substrate, group size/composition, and mixed-species exhibits. Despite these reservations, the manuscript represents an important contribution to the literature and outlines useful recommendations for the behavioral management and exhibit design for this species.
Additional comments:
· It is unfortunate that identification of animals was not possible, making interpretation of some results difficult.
· It would have been helpful to more clearly explain the mixed-species groups within the Methods section. The term “mixed exhibit enclosure” was not clear.
· Figure 1 presents the group composition and enclosure height in an interesting way. However, it would have been helpful to include information about the age and sex of the individual animals, as well as which exhibits contained other species and their number. Including a table in the Supplemental information would be helpful.
· It would have been useful to also provide the actual height dimensions of each zone (1-4) for each of the enclosures.
· The intermingled use of the terms “setting” and “collection” to refer to zoos A-E was a bit confusing.
· There were no data presented regarding inter- or intra-observer reliability.
· It was unclear whether the “scanning” behavioural category actually represented a combination of “deliberate sweeping movement of the head” as well as simple passivity. It represented nearly 60% of the behavioral activity according to Figure 2. Did a significant proportion of that time actually represent scanning or does it represent non-active behaviour instead?
· Figure 4 presents data on intra- and inter-specific grooming. It would be helpful to indicate in the figure that zoos B and E represented mixed-species exhibits while zoos A, C and D represented Callimico-only exhibits.
· Page 8, Line 294 refers to Figure 11. Should this be Figure 5?
· The discussion seems overly long and could be have been made more concise.
Author Response
Thank you for reading our manuscript and your comments which we have answered point by point below.
Additional comments:
- It is unfortunate that identification of animals was not possible, making interpretation of some results difficult.
We were not analysing for individual differences. While some of the collections did have identifiable individuals allowing us to reflect on the interspecific and conspecifc interactions, this was not possible in the larger groups
- It would have been helpful to more clearly explain the mixed-species groups within the Methods section. The term “mixed exhibit enclosure” was not clear.
amended
- Figure 1 presents the group composition and enclosure height in an interesting way. However, it would have been helpful to include information about the age and sex of the individual animals, as well as which exhibits contained other species and their number. Including a table in the Supplemental information would be helpful.
A table has been added to include this
- It would have been useful to also provide the actual height dimensions of each zone (1-4) for each of the enclosures.
This has been included
- The intermingled use of the terms “setting” and “collection” to refer to zoos A-E was a bit confusing.
Setting has been changed
- There were no data presented regarding inter- or intra-observer reliability.
We have edited to clarify that data was collected live by one observer so it is not possible to assess intra-observer reliability
- It was unclear whether the “scanning” behavioural category actually represented a combination of “deliberate sweeping movement of the head” as well as simple passivity. It represented nearly 60% of the behavioral activity according to Figure 2. Did a significant proportion of that time actually represent scanning or does it represent non-active behaviour instead?
We have added a clearer ethogram that was used in the study into the main paper. We hope this helps to answer your question on the definition of scanning.
- Figure 4 presents data on intra- and inter-specific grooming. It would be helpful to indicate in the figure that zoos B and E represented mixed-species exhibits while zoos A, C and D represented Callimico-only exhibits.
This figure has been amended.
- Page 8, Line 294 refers to Figure 11. Should this be Figure 5?
Amended
- The discussion seems overly long and could be have been made more concise.
Thank you for your comment, we have refined the discussion in some places.